# An Algal Metabolite-Based PPAR-γ Agonist Displayed Anti-Inflammatory Effect via Inhibition of the NF-κB Pathway

**DOI:** 10.3390/md17060321

**Published:** 2019-05-30

**Authors:** Zhiran Ju, Mingzhi Su, Dandan Li, Jongki Hong, Dong-Soon Im, Suhkmann Kim, Eun La Kim, Jee H. Jung

**Affiliations:** 1College of Pharmacy, Pusan National University, Busan 46241, Korea; 13719405761@163.com (Z.J.); smz0310@163.com (M.S.); 18840659614@163.com (D.L.); imds@pusan.ac.kr (D.-S.I.); eunlakim@gmail.com (E.L.K.); 2College of Pharmacy, Kyunghee University, Seoul 02447, Korea; jhong@khu.ac.kr; 3Center for Proteome Biophysics, Department of Chemistry, Pusan National University, Busan 46241, Korea; suhkmann@pusan.ac.kr

**Keywords:** PPAR-γ agonist, 15d-PGJ_2_, anti-inflammatory, NF-κB pathway

## Abstract

In our previous study, a synthetic compound, (+)-(*R*,*E*)-**6a1**, that incorporated the key structures of anti-inflammatory algal metabolites and the endogenous peroxisome proliferator-activated receptor γ (PPAR-γ) ligand 15-deoxy-∆^12,14^-prostaglandin J_2_ (15d-PGJ_2_), exerted significant PPAR-γ transcriptional activity. Because PPAR-γ expressed in macrophages has been postulated as a negative regulator of inflammation, this study was designed to investigate the anti-inflammatory effect of the PPAR-γ agonist, (+)-(*R*,*E*)-**6a1**. Compound (+)-(*R*,*E*)-**6a1** displayed in vitro anti-inflammatory activity in lipopolysaccharides (LPS)-stimulated murine RAW264.7 macrophages. Compound (+)-(*R*,*E*)-**6a1** suppressed the expression of proinflammatory factors, such as nitric oxide (NO), inducible NO synthase (iNOS), cyclooxygenase-2 (COX-2), interleukin-6 (IL-6), and tumor necrosis factor-α (TNF-α), possibly by the inhibition of the nuclear factor-κB (NF-κB) pathway. In macrophages, (+)-(*R*,*E*)-**6a1** suppressed LPS-induced phosphorylation of NF-κB, inhibitor of NF-κB α (IκBα), and IκB kinase (IKK). These results indicated that PPAR-γ agonist, (+)-(*R*,*E*)-**6a1,** exerts anti-inflammatory activity via inhibition of the NF-κB pathway.

## 1. Introduction

Peroxisome proliferator-activated receptor γ (PPAR-γ), which can be activated by natural or synthesized ligands, such as 15-deoxy-∆^12,14^-prostaglandin J_2_ (15d-PGJ_2_) or rosiglitazone, respectively, is a member of the nuclear receptor superfamily [1]. Generally, inactive PPAR-γ may be localized to the cytoplasm rather than the nucleus. Activated PPAR-γ is translocated to the cell nucleus and forms a heterodimer with a second member of the nuclear receptor family, retinoic X receptor (RXR). Then, the PPAR/RXR heterodimer binds to peroxisome proliferator response element (PPRE) on target DNA to regulate the transcription of genes relevant to lipid and glucose metabolism [2]. Recent research also revealed that PPAR-γ plays a key role in the repression of inflammatory genes, especially in macrophages. PPAR-γ expressed in macrophages has been postulated as a negative regulator of inflammation [3,4]. After activation by infection, tissue damage, or exposure to endotoxin (i.e., lipopolysaccharides (LPS)), macrophages secrete a large amount of proinflammatory mediators, including inducible nitric oxide synthase (iNOS), nitric oxide (NO), cyclooxygenase-2 (COX-2), tumor necrosis factor-α (TNF-α), and interleukin-6 (IL-6). Some of these proinflammatory mediators are involved in systemic diseases, such as obesity, diabetes, and cancer [5,6,7]. Currently, two molecular mechanisms have been defined for anti-inflammatory actions of PPAR-γ agonist. (1) PPAR-γ agonist binds to NF-κB (nuclear factor-κB) in the nucleus to inhibit its binding to DNA gene promoter regions, resulting in the suppression of inflammatory gene transcription. Binding of active NF-κB to DNA leads to the expression of proinflammatory mediators, such as TNF-α, IL-1β, IL-6, iNOS, and COX-2 [8,9,10,11,12,13,14]. (2) PPAR-γ agonist, such as 15d-PGJ_2_, interferes with activation of NF-κB in the cytoplasm. NF-κB resides in the cytoplasm under inactive state while associated with its repressor, IκB (inhibitor of NF-κB). During inflammatory stimulation, IκB may undergo phosphorylation by IκB kinase (IKK) and, thus, be degraded, leading to the release of NF-κB and allowing it to enter the nucleus. The endogenous PPAR-γ ligand, 15d-PGJ_2_, was shown to covalently bind to IKK, inhibiting its function and subsequently inhibiting the phosphorylation of IκB and then activation of NF-κB [8,9,10,11,12,13].

In a previous study, we isolated oxy fatty acids and prostaglandins from a red alga with substantial anti-inflammatory activity [15]. Thereafter, by incorporating the key structural motifs of these natural products, we synthesized a new class of endocyclic enone jasmonate derivatives as anti-inflammatory leads [16,17]. In a further study, we synthesized an exocyclic enone jasmonate derivative, (+)-(*R*,*E*)-**6a1**, as a potent PPAR-γ agonist that share the key exocyclic enone moiety with the endogenous PPAR-γ ligand, 15d-PGJ_2_ [18] (See Appendix A). Typical PPAR-γ ligands, such as rosiglitazone and 15d-PGJ_2_, are composed of three distinct partial structures, a polar head, linker, and a hydrophobic tail (Figure 1). The polar head and hydrophobic tail play important roles in H-bonding and hydrophobic interaction, respectively, with the PPAR-γ LBD (ligand binding domain). The exocyclic *α,β*-unsaturated ketone (enone) moiety of 15d-PGJ_2_ was reported to be essential for covalent bonding with Cys^285^ in the PPAR-γ LBD. Similar to 15d-PGJ_2_, the enone functionality of (+)-(*R*,*E*)-**6a1** may contribute to covalent bonding to the PPAR-γ LBD, and this additional covalent bonding may contribute to the activation of PPAR-γ [18,19,20]. In a continuing study, we investigated the in vitro anti-inflammatory effects of the PPAR-γ agonist, (+)-(*R*,*E*)-**6a1,** in RAW264.7 murine macrophages. Based on in vitro results, the possible anti-inflammatory mechanism of (+)-(*R*,*E*)-**6a1** was also discussed.

## 2. Results and Discussion

### 2.1. Compound (+)-(R,E)-6a1 Promoted PPAR-γ Translocation to Cell Nuclei

Generally, after ligand binding, the activated PPAR-γ will translocate to the nucleus and bind to NF-κB to repress the gene expression of proinflammatory mediators. In our previous study, the PPAR-γ agonistic activity of (+)-(*R*,*E*)-**6a1** was evaluated by luciferase assay by using the PPRE-luciferase reporter plasmid [18]. Herein, we used Western blot to assess the protein level of the translocated PPAR-γ at the nucleus due to activation by (+)-(*R*,*E*)-**6a1** in RAW264.7 cells. The result showed that the endonuclear PPAR-γ protein level was significantly increased by (+)-(*R*,*E*)-**6a1** treatment in a concentration-dependent manner, and the activity was comparable to the standard PPAR-γ agonist rosiglitazone (Figure 2). Since PPAR-γ expressed in macrophages can downregulate inflammatory responses [21], we investigated the expression of proinflammatory mediators.

### 2.2. Cytotoxicity of (+)-(R,E)-6a1 to RAW264.7, Ac2F, and KB Cells

Prior to the in vitro anti-inflammatory assay, compound (+)-(*R*,*E*)-**6a1** was first evaluated for its cytotoxicity to murine macrophages (RAW264.7), rat liver cells (Ac2F), and human oral epidermoid cancer cells (KB) to gauge the suitable concentration for the cell-based anti-inflammatory assay. As shown in Figure 3, (+)-(*R*,*E*)-**6a1** showed no significant toxicity to cells at the concentrations of 10 and 50 μM for 24 h; specifically, (+)-(*R*,*E*)-**6a1** was nontoxic to RAW264.7 cells at the concentration of 50 μM for up to 24 h. However, the viability and cell morphology of RAW264.7 cells appeared to be affected by (+)-(*R*,*E*)-**6a1** after 48 h at 50 μM (Figure 3D). Therefore, RAW264.7 cells were employed for anti-inflammatory assay with (+)-(*R*,*E*)-**6a1** concentrations lower than 30 μM.

### 2.3. (+)-(R,E)-6a1 Inhibited LPS-Induced Expression of Proinflammatory Factors in RAW264.7 Cells

In order to verify the in vitro anti-inflammatory effect of (+)-(*R*,*E*)-**6a1**, the protein levels of the proinflammatory factors, iNOS and COX-2, were examined in RAW264.7 cells by Western blot. As expected, LPS stimulation markedly increased iNOS and COX-2 protein levels, but this increase could be diminished in a dose-dependent manner by pretreatment with (+)-(*R*,*E*)-**6a1** (Figure 4). Notably, at the concentration of 30 μM, (+)-(*R*,*E*)-**6a1** significantly decreased the protein levels of iNOS and COX-2 with a potency comparable to that of 10 μM dexamethasone. Dexamethasone was employed as a standard anti-inflammatory drug for comparison.

Meanwhile, as an inflammatory mediator, high levels of NO are produced in response to inflammatory stimuli and mediation of inflammatory effects. iNOS is a family of enzymes that catalyze NO production from L-arginine. Thereby, it was hypothesized that the suppression of iNOS by (+)-(*R*,*E*)-**6a1** (Figure 4) would lead to decreased NO production in macrophages, and thus, we examined NO levels in RAW264.7 cell supernatant using the Griess reagent. Compound (+)-(*R*,*E*)-**6a1** significantly decreased the NO production in a concentration-dependent manner (Figure 5A). In addition, the amount of TNF-α and IL-6 that was produced was analyzed by enzyme-linked immunosorbent assay (ELISA). TNF-α and IL-6 are inflammatory cytokines that provide a host of defensive effects during the inflammatory response and maintain normal cellular conditions [22]. Proinflammatory mediator levels were markedly increased when murine macrophages RAW264.7 were exposed to LPS. However, these increases in TNF-α and IL-6 were inhibited by (+)-(*R*,*E*)-**6a1** in a dose-dependent manner (Figure 5B,C), suggesting that (+)-(*R*,*E*)-**6a1** participates in a signaling pathway activated by LPS in macrophages. This result is consistent with that of other studies, which have shown that treatment of macrophages with various concentrations of PPAR-γ agonists reduces the production of proinflammatory cytokines [23,24].

### 2.4. (+)-(R,E)-6a1 Inhibited LPS-Induced NF-Κb Signal Pathway in RAW264.7 Cells

NF-κB is recognized as a crucial component of many immune responses and inflammation. For example, macrophages rely on NF-κB for the secretion of proinflammatory cytokines [25]. The conventional anti-inflammatory mechanism of PPAR-γ ligands is known via the inhibition of NF-κB [8,9,10,11,12,13,14]. PPAR-γ ligands, such as rosiglitazone, activate PPAR-γ, and activated PPAR-γ is translocated into the nucleus to bind with NF-κB. The NF-κB–PPAR-γ complex cannot bind to the promotor region of DNA; thereby, gene expression of proinflammatory mediators is suppressed (Figure 6, path A). Meanwhile, an endogenous PPAR-γ ligand, 15-deoxy-∆^12,14^-prostaglandin J_2_ (15d-PGJ_2_), was reported to exert anti-inflammatory activity via additional mechanism by inhibiting the activation and nuclear translocation of NF-κB (Figure 6, path B). Recently, Rossi et al. indicated that IKK is the critical target of 15d-PGJ_2_ in the NF-κB activation pathway. The study showed that the exocyclic enone moiety in 15d-PGJ_2_ can form Michael adducts that covalently modify Cys^179^ of IKK, thus leading to the inhibition of IKK phosphorylation, and subsequent interference with the downstream NF-κB activation events [9,11].

Since (+)-(*R,E*)-**6a1** share the same exocyclic enone moiety with 15d-PGJ_2_, we investigated the effect of (+)-(*R,E*)-**6a1** on the alternative pathway of 15d-PGJ_2_ involving the inhibition of IKK and subsequent inhibition of NF-κB. As shown in Figure 7, the phosphorylation levels of NF-κB significantly increased after LPS treatment for 30 min, but pretreatment with (+)-(*R*,*E*)-**6a1** obviously decreased the NF-κB p65 phosphorylation in a dose-dependent manner (Figure 7A). As a result, the phosphorylated protein level of NF-κB in the nucleus was also significantly decreased by (+)-(*R*,*E*)-**6a1** treatment (Figure 7B). IKK and IκB activate NF-κB, because phosphorylated IκBα by IKK releases the active NF-κB for translocation into the nucleus [26]. As expected, LPS stimulation markedly increased phosphorylated protein levels of IKK and IκBα. However, the phosphorylation of IKK and IκBα was decreased (Figure 7C,D), and IκBα degradation was prevented in a dose-dependent manner (Figure 7E) by (+)-(*R*,*E*)-**6a1** treatment. At the same time, the immunofluorescence assay showed that the LPS-stimulated translocation of NF-κB into the nucleus was moderately prevented by (+)-(*R*,*E*)-**6a1**, especially at the concentration of 30 μM (Figure 7F). Our findings suggested that (+)-(*R*,*E*)-**6a1** may exert anti-inflammatory activity not only by the conventional inhibition of NF-κB from DNA binding like typical PPAR-γ ligands (Figure 6, path A), but also by inhibition of the activation and endonuclear translocation of NF-κB (Figure 6, path B) in the same manner as the 15d-PGJ_2_, which shares the same exocyclic enone moiety.

## 3. Materials and Methods

### 3.1. Materials

Compound (+)-(*R*,*E*)-**6a1** was synthesized by our group [18]. Dimethylsulfoxide (DMSO), dexamethasone (DEX), lipopolysaccharide (LPS), Griess reagent were purchased from Sigma-Aldrich (St. Louis, MO, USA). NE-PER^TM^ Nuclear and Cytoplasmic Extraction Reagents kit was purchased from Thermo Scientific (Rockford, IL, USA).

### 3.2. Cell Culture and Cell Viability

RAW264.7 murine macrophages were purchased from the Korean Cell Line Bank (KCLB^®^, Seoul, Korea); rat liver Ac2F cells and human oral epidermoid cancer cells (KB) were obtained from the American Type Culture Collection (ATCC, Rockville, MD, USA). Cells were cultured at 37 °C in a 5% CO_2_ humidified incubator and maintained in high-glucose Dulbecco’s Modified Eagle Medium (DMEM, Nissui, Tokyo, Japan) containing 100 mg/mL streptomycin, 2.5 mg/L amphotericin B, and 10% heat-inactivated fetal bovine serum (FBS). Suspensions of tested cell lines (cal. 1.0 × 10^4^ cells/well) were seeded in 96-well culture plates, cultured for 12 h, and then treated with various diluted concentrations of (+)-(*R*,*E*)-**6a1** for 24 h, 48 h, and 72 h, respectively. Control cultures were treated with culture medium alone. The tested compounds were evaluated at twice-fold dilutions, and the highest concentration was 50 μM. Cell viability was evaluated using water soluble tetrazolium (WST) reagent (EZ-CyTox, Daeil Lab Service Co., Ltd., Seoul, Korea), which was added to each well (10 μL) and incubated at 37 °C for 1 h. Absorbances were read using an iMark Microplate Absorbance Reader (Bio-Rad Laboratories, Hercules, CA, USA) at a wavelength of 450 nm. Cells in the exponential phase were used for all experiments.

### 3.3. Production Levels of NO and Cytokines Released into the Medium

RAW264.7 macrophages (cal. 1 × 10^4^ cells/well) were seeded in a 96-well culture plate and cultured for 12 h. Cells were pretreated with various concentrations of drug for 1 h and then co-incubated with 25 ng/mL of LPS for 24 h. NO concentrations in medium were determined using a Griess assay. Griess reagent (80 μL) was added to media supernatants (80 μL) and then incubated at 37 °C for 15 min in the dark. Absorbance was measured at 520 nm using an iMark Microplate Absorbance Reader (Bio-Rad Laboratories, Hercules, CA, USA). NO concentrations were calculated using 0–100 μM sodium nitrite standards. TNF-α and IL-6 expression levels in culture medium were quantified using a sandwich-type ELISA kit (Biolegend, San Diego, CA, USA). Absorbance was measured at 450 nm.

### 3.4. Immunofluorescence Staining of NF-Κb P65 in RAW264.7 Cells

Cells were grown on confocal dish and treated with compound treatment for 24 h. After treatment, cells were fixed in 10% formalin solution for 15 min, washed with phosphate buffer saline (PBS) thrice, treated with 0.5% (v/v) Triton X-100/PBS for 15 min, washed with PBS thrice, and then blocked at room temperature for 30 min in 10% FBS/PBS. Cells were incubated with rabbit anti-NFκB-p65 antibody (Cell signaling technology, Danvers, MA, USA) at 4 °C overnight, washed thrice with PBS, incubated for 30 min at room temperature with secondary antibody anti-rabbit Alexa 488 (Cell signaling technology, USA) as a molecular probe, washed thrice with PBS, and then incubated with PI/Rnase (10 µg/mL) at room temperature for 20 min. The location of NFκB-p65 was viewed with a confocal microscopy FluoView FV10i (Olympus, Australia) using an excitation wavelength of 488 nm and an emission wavelength of 537 nm.

### 3.5. Western Blot Assay

RAW264.7 cells were harvested and suspended in lysis buffer containing protease and phosphatase inhibitor cocktails. The concentration of proteins was determined using a bicinchoninic acid (BCA) protein assay (Thermo Scientific, Rockford, IL, USA). Equal amounts of proteins were resolved by 10% sodium dodecyl sulfate (SDS)-polyacrylamide gel electrophoresis and electrophoretically transferred to polyvinylidene difluoride (PVDF) membranes, which were then blocked in Tris-buffered saline containing 0.1% Tween 20 (TBS-T) and 5% skimmed milk for 1 h at room temperature. Then, the membranes were incubated with specific primary antibodies (Cell Signaling Technology, Danvers, MA, USA) overnight at 4 °C. Anti-rabbit IgG-HRP was used as the secondary antibody. Signals were developed using the ChemiDoc™Touch Imaging System (Bio-Rad Laboratories, Hercules, CA, USA).

### 3.6. Statistical Analysis

The significance of intergroup differences was determined by ANOVA. Results are expressed as the mean ± SDs of indicated numbers of independent experiments. Values of *p* < 0.05 were considered statistically significant.

## 4. Conclusions

To summarize, the in vitro anti-inflammatory activity of PPAR-γ ligand, (+)-(*R*,*E*)-**6a1,** was evaluated, and the anti-inflammatory mechanism of (+)-(*R*,*E*)-**6a1** was discussed. Compound (+)-(*R*,*E*)-**6a1** decreased the protein levels of iNOS and COX-2 with a potency comparable to that of dexamethasone. The productions of NO, IL-6, and TNF-α were decreased by (+)-(*R*,*E*)-**6a1** by a similar degree as dexamethasone. Compound (+)-(*R*,*E*)-**6a1** suppressed LPS-induced phosphorylation of NF-κB, IKK, and IκBα in macrophages. Collected data indicated that (+)-(*R*,*E*)-**6a1** may exert anti-inflammatory effect in a similar way as an endogenous PPAR-γ ligand, 15d-PGJ_2_, via two distinct NF-κB inhibition pathways. The results of the in vitro anti-inflammatory potency of (+)-(*R*,*E*)-**6a1**, which is quite comparable to that of dexamethasone, suggest that (+)-(*R*,*E*)-**6a1** may serve as a potential anti-inflammatory lead for further study.

## Figures and Tables

**Figure 1 marinedrugs-17-00321-f001:**
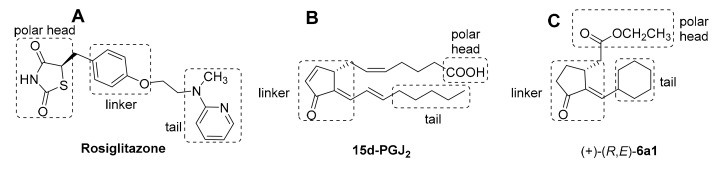
Structures and pharmacophoric regions of (**A**) rosiglitazone, (**B**) 15-deoxy-∆^12,14^ prostaglandin J_2_ (15d-PGJ_2_), and (**C**) (+)-(*R*,*E*)-**6a1**.

**Figure 2 marinedrugs-17-00321-f002:**
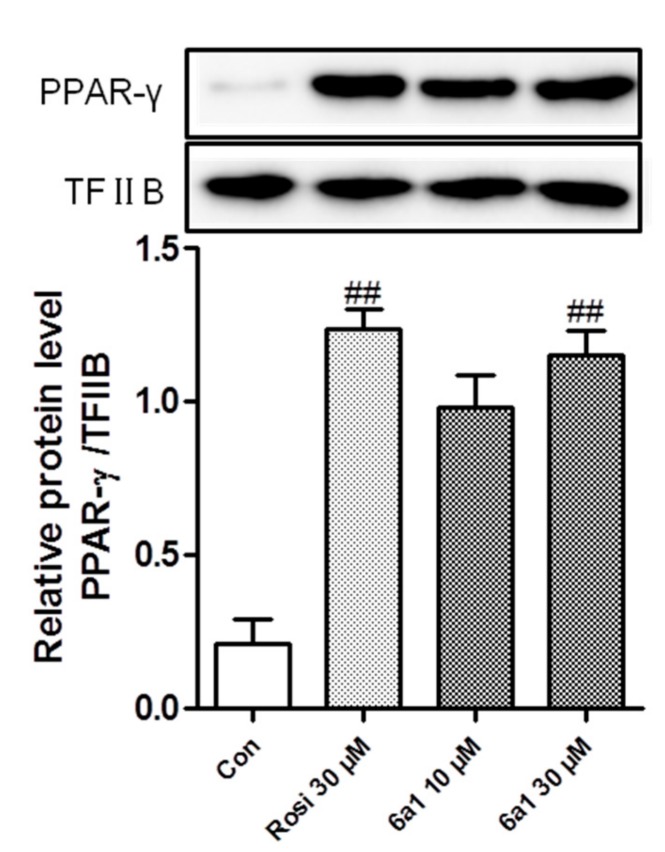
Endonuclear peroxisome proliferator-activated receptor γ (PPAR-γ) protein levels in RAW264.7 macrophages treated with (+)-(*R*,*E*)-**6a1** for 24 h were determined by Western blotting. Nuclear levels of transcription factor II B (TF II B) were used for reference purposes. Rosiglitazone (Rosi) was employed as a positive control. The results shown are representative of three independent experiments. ^##^
*p* < 0.01 compared with the control group.

**Figure 3 marinedrugs-17-00321-f003:**
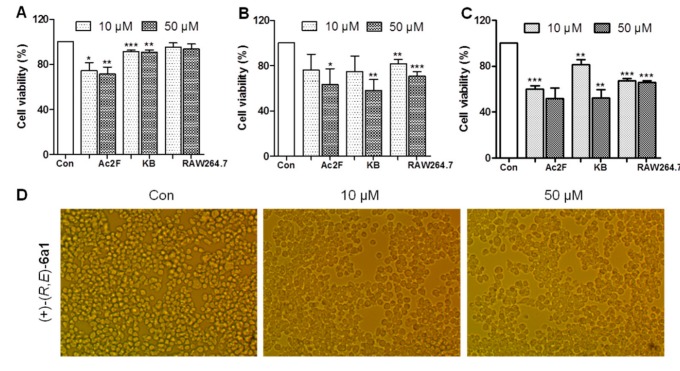
Cytotoxicity of (+)-(*R*,*E*)-**6a1** on rat liver Ac2F cells, human oral epidermoid cancer cells (KB), and RAW 264.7 murine macrophages at (**A**) 24 h, (**B**) 48 h, and (**C**) 72 h. (**D**) The morphological changes of RAW264.7 cells following treatment with (+)-(*R*,*E*)-**6a1** for 48 h. Magnification 100×. The results are shown as mean ± SD (n = 3) of three independent experiments. * *p* < 0.05, ** *p* < 0.01, *** *p* < 0.001 compared with the control group.

**Figure 4 marinedrugs-17-00321-f004:**
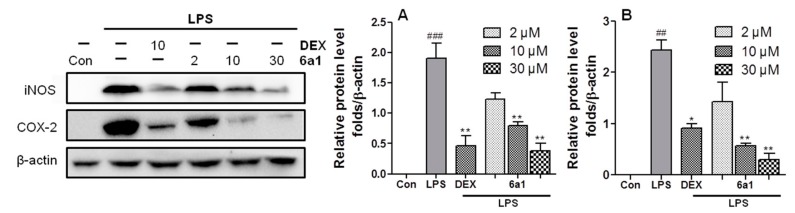
Effects of (+)-(*R*,*E*)-**6a1** on liposaccharides (LPS)-induced (**A**) nitric oxide synthase (iNOS) and (**B**) cyclooxygenase-2 (COX-2) protein expression in RAW264.7 cells. The cells were treated with different concentrations of (+)-(*R*,*E*)-**6a1**, and cultured in the presence or absence of LPS (1 μg/mL) for 24 h. Dexamethasone (DEX) was employed as a positive control (10 μM). The results are shown as mean ± SD (n = 3) of three independent experiments. ^##^
*p* < 0.01, ^###^
*p* < 0.001 compared with the control group; * *p* < 0.05, ** *p* < 0.01 compared with the LPS-stimulated group.

**Figure 5 marinedrugs-17-00321-f005:**
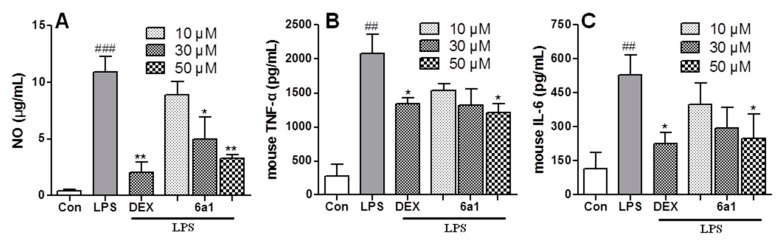
Inhibitory activity of (+)-(*R*,*E*)-**6a1** on lipopolysaccharides (LPS)-induced nitric oxide (NO) and cytokine production. (**A**) The production of NO in the medium of RAW264.7 cells after treatment with (+)-(*R*,*E*)-**6a1** for 1 h followed by treatment with LPS (25 ng/mL, 24 h). The concentration of NO in the medium was determined by Griess method. (**B**) The production of tumor necrosis factor α (TNF-α) in the RAW264.7 cells after treatment with (+)-(*R*,*E*)-**6a1** for 1 h, and then with LPS (25 ng/mL, 3 h). (**C**) The production of interleukin-6 (IL-6) in the medium of RAW264.7 cells after treatment with (+)-(*R*,*E*)-**6a1** for 1 h, and then with LPS (25 ng/mL, 24 h). Dexamethasone (DEX) was employed as a positive control (30 μM). The results are shown as mean ± SD (n = 3) of three independent experiments. ^##^
*p* < 0.01, ^###^
*p* < 0.001 compared with the control group; * *p* < 0.05, ** *p* < 0.01 compared with the LPS-stimulated group.

**Figure 6 marinedrugs-17-00321-f006:**
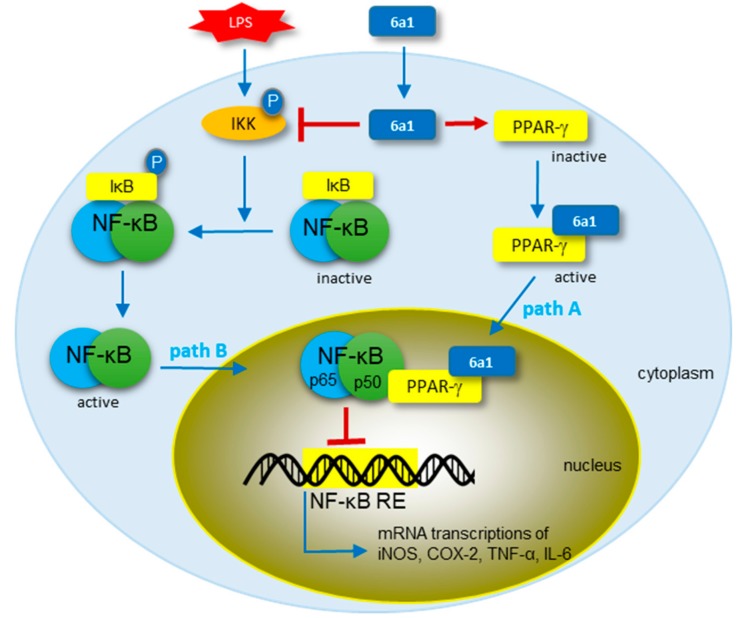
The speculative anti-inflammatory mechanism of (+)-(*R*,*E*)-**6a1** on the nuclear factor-κB (NF-κB) signal pathway in RAW 264.7 cells. Path A: Conventional anti-inflammatory mechanism of peroxisome proliferator-activated receptor γ (PPAR-γ) ligands via binding with NF-κB in the nucleus, and thereby blocking it from binding to DNA promotor regions. Path B: Alternative anti-inflammatory mechanism of 15d-PGJ_2_ via the inhibition of NF-κB activation and endonuclear translocation.

**Figure 7 marinedrugs-17-00321-f007:**
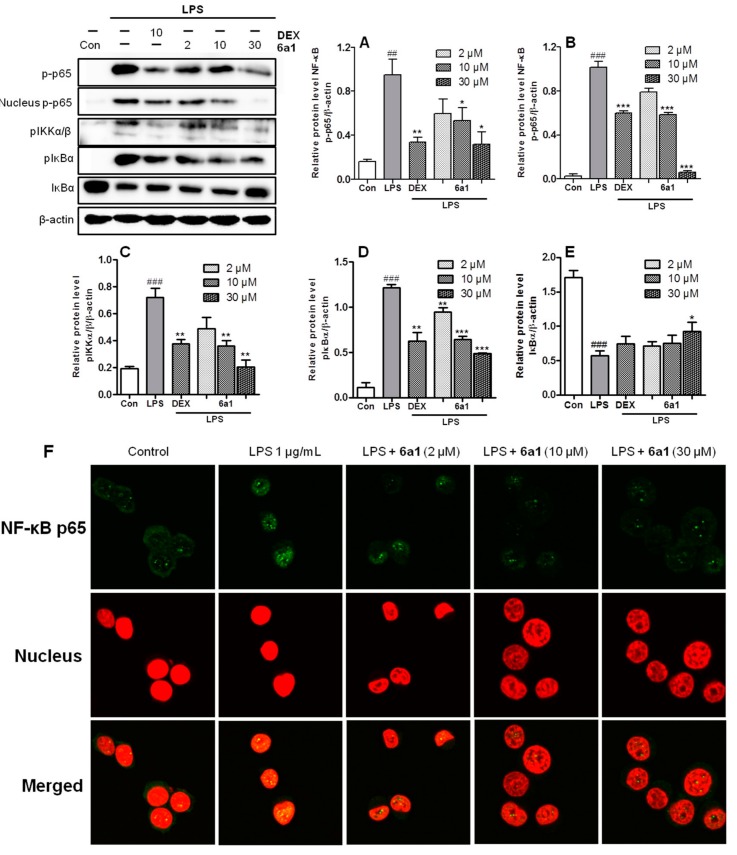
Effect of compound (+)-(*R*,*E*)-**6a1** on the nuclear factor-κB (NF-κB) p65 activation in RAW 264.7 macrophages. (**A**) Phosphorylation of NF-κB p65, (**B**) phosphorylated endonuclear NF-κB p65, (**C**) phosphorylation of IκB kinase (IKK), (**D**) phosphorylation of IκBα, and (**E**) IκBα protein were determined by Western blot. Cells were pretreated with compound (+)-(*R*,*E*)-**6a1** for 1 h, and then stimulated with lipopolysaccharide (LPS, 25 ng/mL) for 30 min. β-actin was used as an internal control. (**F**) Immunofluorescence assay, NF-κB p65 is viewed as green fluorescence, and cell nuclei are viewed as red fluorescence by PI staining using confocal microscopy. The results shown are representative of three independent experiments. ^##^
*p* < 0.01, ^###^
*p* < 0.001 compared with the control group; * *p* < 0.05, ** *p* < 0.01, *** *p* < 0.001 compared with the LPS-stimulated group.

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
