# Peer review of "An Algal Metabolite-Based PPAR-γ Agonist Displayed Anti-Inflammatory Effect via Inhibition of the NF-κB Pathway"

_marinedrugs, 2019, doi:10.3390/md17060321_

Round 1
Reviewer 1 Report
This manuscript, marinedrugs-513389, was described that PPAR-γ agonist, (+)-(R,E)-6a1, exerts anti-inflammatory activity via inhibition of the NF-κB pathway. This manuscript was interesting that PPAR-γ level of the author's original synthetic compound was elaborated for support of their biological mechanism. Thus, this manuscript is recommended to publish in Marine Drug after minor revision.
Minor revision,
1) In Figure 2 and 3, positive controls were different. Why did the author choose two different type of compound. Please discuss.
2) Comparing between 6a1, 15d-PGJ2, and dexamethasone, which is compound highest potential? Please discuss.
Author Response
This manuscript, marinedrugs-513389, was described that PPAR-γ agonist, (+)-(R,E)-6a1, exerts anti-inflammatory activity via inhibition of the NF-κB pathway. This manuscript was interesting that PPAR-γ level of the author's original synthetic compound was elaborated for support of their biological mechanism. Thus, this manuscript is recommended to publish in Marine Drug after minor revision.
Minor revision,
1) In Figure 2 and 3, positive controls were different. Why did the author choose two different type of compound. Please discuss.
Answer: Thank you for your valuable comments. I guess the reviewer is referring to Figures 2 and 4. In Figure 2, we evaluated PPAR-γ transactivation level. Since rosiglitazone is the typical and clinical PPAR-γ agonist, it was used as the positive control. In Figure 4, we evaluated the in vitro anti-inflammatory activity. Since dexamethasone is the clinical drug for treatment of inflammatory symptoms, we used dexamethasone as positive control.
As suggested, we included sentences as follows.
Lines 78-79: …….. and the activity was comparable to the standard PPAR-γ agonist rosiglitazone (Figure 2).
Lines 108-109: ….. Dexamethasone was employed as a standard anti-inflammatory drug for comparison.
2) Comparing between 6a1, 15d-PGJ2, and dexamethasone, which is compound highest potential? Please discuss.
Answer: Thank you for your invaluable comment. The main topic of this paper was evaluation and comparison of the in vitro anti-inflammatory activity of (+)-(R,E)-6a1. Anti-inflammatory activity of (+)-(R,E)-6a1 (10 μΜ) was compared with dexamethasone (10 μΜ), and they were almost eqipotent. Therefore, we included a sentence as suggested.
Line 258: …., which is quite comparable to that of dexamethasone,
Reviewer 2 Report
An Algal Metabolite-based PPAR-γ Agonist Displayed Anti-Inflammatory Effect via Inhibition of the NF-κB Pathway
Zhiran Ju, Mingzhi Su, Dandan Li, Jongki Hong, Dong-Soon Im, Suhkmann Kim, Eun La Kim , and Jee H. Jung
This is a very well written paper, concsie and up to the point. The experimental designs are well thought of and all the data presented neatly. General presentation excellent and the write up clear.
In figure 4 legend and to be consistent with the other figures the time scale of drug (6a1) pre-treatment can be included.
Thoroughly enjoyed reading the paper.
Author Response
This is a very well written paper, concise and up to the point. The experimental designs are well thought of and all the data presented neatly. General presentation excellent and the write up clear.
In figure 4 legend and to be consistent with the other figures the time scale of drug (6a1) pre-treatment can be included.
Thoroughly enjoyed reading the paper.
Answer: Thank you for your invaluable comment. Actually the cells were cultured in the presence of (+)-(R,E)-6a1 for 24 h. To avoid any obscure description, the sentence was changed as follows.
The cells were treated with different concentrations of (+)-(R,E)-6a1, and then cultured in the presence or absence of LPS (1 μg/mL) for 24 h.
The cells were treated with different concentrations of (+)-(R,E)-6a1, and cultured in the presence or absence of LPS (1 μg/mL) for 24 h (removed ‘then”)